# Dual-Branch Network with Hybrid Attention for Multimodal Ophthalmic Diagnosis

**DOI:** 10.3390/bioengineering12060565

**Published:** 2025-05-25

**Authors:** Xudong Wang, Anyu Cao, Caiye Fan, Zuoping Tan, Yuanyuan Wang

**Affiliations:** 1School of Ophthalmology & Optometry, Wenzhou Medical University, Wenzhou 325000, China; xudong.wang.apply@outlook.com (X.W.); anyu_cao@163.com (A.C.); 2School of Data Science and Artificial Intelligence, Wenzhou University of Technology, Wenzhou 325000, China; caiyef@aliyun.com

**Keywords:** multimodal, hybrid attention mechanisms, classification of eye diseases, dual-branch learning network, imbalanced data

## Abstract

In this paper, we propose a deep learning model based on dual-branch learning with a hybrid attention mechanism for meeting challenges in the underutilization of features in ophthalmic image diagnosis and the limited generalization ability of traditional single modal deep learning models when using imbalanced data. Firstly, a dual-branch architecture layout is designed, in which the left and right branches use residual blocks to deal with the features of a 2D image and 3D volume, respectively. Secondly, a frequency domain transform-driven hybrid attention module is innovated, which consists of frequency domain attention, spatial attention, and channel attention, respectively, to solve the problem of inefficiency in network feature extraction. Finally, through a multi-scale grouped attention fusion mechanism, the local details and global structure information of the bimodal modalities are integrated, which solves the problem of the inefficiency of fusion caused by the heterogeneity of modal features. The experimental results show that the accuracy of MOD-Net improved by 1.66% and 1.14% over GeCoM-Net and ViT-2SPN, respectively. It can be concluded that the model effectively mines the deep correlation features of multimodal images through the hybrid attention mechanism, which provides a new paradigm for the intelligent diagnosis of ophthalmic diseases.

## 1. Introduction

The rapid development of artificial intelligence (AI) has dramatically changed various fields, including healthcare [1]. In the field of ophthalmology, AI technologies are increasingly being integrated into clinical practice, especially for the early detection and diagnosis of ophthalmic diseases [2]. For example, diseases such as diabetic retinopathy, age-related macular degeneration, and glaucoma pose a great threat to human beings due to their high prevalence and potential loss of vision. Among them, glaucoma is a chronic neurodegenerative disease characterized by optic nerve damage and visual field defects, which ultimately leads to irreversible blindness and is usually caused by high intraocular pressure (IOP), which can be clinically observed as a large cup-to-disc ratio on fundus color photographs [3]. Age-related macular degeneration (AMD) is a degenerative eye disease that primarily affects the vision of the elderly and is characterized by a progressive loss of central vision, the incidence of which increases significantly with age [4]. Diabetic retinopathy (DR) is a common complication in diabetic patients, which is caused by hyperglycemia and leads to retinal microvascular damage, triggering vision loss and even blindness [5]. However, artificial intelligence-enhanced retinal imaging enables clinicians to identify biomarkers for systemic diseases [6].

Currently, deep learning is widely used in fundus image analysis and retinopathy detection for automatically processing ophthalmic data and detecting features that may indicate disease [7], offering the possibility of enhancing the diagnostic process and improving patient prognosis. Atrophic age-related macular degeneration can be predicted using an algorithm for atrophy progression based on deep learning techniques [8]. Deep learning has also been applied to ophthalmic imaging for the early detection of neurodegenerative diseases [9]. P. Racioppo et al. [10] used knowledge distillation to perform FAZ segmentation on OCTA 2D en-face images. One of the major applications of deep learning in ophthalmology is the automated detection of diabetic retinopathy [11], which is the main cause of vision loss in diabetic patients. G. Lim et al. [12] proposed a VGGNet-based convolutional neural network for the combined detection of three eye diseases (DR/AMD/glaucoma), which would reduce the burden on ophthalmologists and facilitate early intervention. Deep learning has also been applied to the diagnosis and early screening of glaucoma. H. Fu et al. [13] proposed a novel disc-aware ensemble network (DENet) for automatic glaucoma screening, which accurately segmented the optic disc and optic cup in color photographs of the fundus to improve the accuracy of glaucoma screening. Deep learning has also been used for ophthalmic surgical assistance and has demonstrated its potential in ophthalmic surgical assistive systems. Negin Ghamsarian et al. [14] proposed a semantic segmentation technique based on a pyramid view and deformable pyramid reception for cataract surgery. More recently, Mohammadreza Saraei et al. [15] proposed a vision transformer-based dual-stream network for OCT classification.

However, despite the fact that deep learning technology has fulfilled certain applications in the field of ophthalmology, there are still some key challenges. First, with the development of technology, its limitations in clinical applications are gradually being revealed. Deep learning models usually require a large amount of labeled data for training [16]; in the field of ophthalmology, on the one hand, the number of datasets is more limited compared to other organs, and on the other hand, obtaining high-quality labeled data often requires interdisciplinary cooperation, which is a challenging task. Secondly, images acquired using different devices and under different imaging conditions may differ significantly, and the generalization ability of the model to different datasets is limited, leading to increased difficulty in traditional model training. Finally, the diversity and complexity of ophthalmic data is an issue, as most of the current studies focus only on the analysis of 2D or 3D modal data [17], as shown in Figure 1, which can only extract and analyze unimodal image features and do not make full use of the potential value of multimodal data in ophthalmology. In fact, the diagnosis and treatment of ophthalmic diseases involve multiple imaging techniques, such as OCTA, OCT, and fundus color photographs, and the analysis of a single modality alone may lead to less comprehensive information acquisition, thereby affecting the accuracy of the diagnostic model. For example, other pathologies (e.g., pathological changes such as high myopia) may affect the accuracy of deep learning models in glaucoma assessment [18]. The authors of [19] have shown that synthesizing information from multiple modalities can significantly improve the accuracy of ocular disease detection.

Therefore, in this paper, the application of multimodal-based deep learning methods in the diagnosis and classification of ophthalmic diseases is deeply investigated, and a dual-branch multimodal network MOD-Net based on frequency-domain transformed attention is constructed, which extracts the features of ocular 2D and 3D modal data and achieves a more comprehensive and accurate analysis of ocular tissues through feature fusion while using information on planar details and stereo structures. By integrating the feature information of different modalities, the robustness of the model and the diagnostic accuracy of the disease are improved, and new ideas and methods are also provided for the development of ophthalmic artificial intelligence. The contributions of this study are as follows:

The MOD-Net is proposed, and experiments on two ophthalmic multimodal datasets, GAMMA and OCTA-500, show that the multimodal nature of the MOD-Net achieves better classification performance for ophthalmic disease diagnosis on small-sample datasets with multiclass imbalance, compared with traditional single-branch networks.A feature extraction optimization process, based on a hybrid attention module FTCA with non-local receptive fields and cross-scale fusion within the convolutional units, is proposed to optimize the feature extraction over the channel and the spatial information of the branching network so that it can better capture the key features of the image.Dual-feature fusion (DFF), a multi-attention mechanism module for 2D and 3D modal feature fusion, is proposed that effectively improves the model’s ability to characterize data features, achieves better feature fusion performance between different modalities, and further improves the image classification and diagnosis performance.Using migration learning and data augmentation, pre-trained weight parameters are migrated into the feature extraction branch network, thereby accelerating the training process and improving the overall generalization ability of the model.

## 2. Related Work

Multimodal learning utilizes data from different modalities for joint learning to improve the performance of models, and multimodal fusion is the core of multimodal learning [20]. Multimodal image fusion techniques have received attention in the field of ophthalmic AI diagnosis and treatment. Y. Wang et al. [21] proposed GeCoM-Net to improve the automatic diagnosis of retinal diseases by combining information from fundus color photographs and optical coherence tomography (OCT) images. However, there are problems with the robustness of modal alignment and an insufficient ability to handle missing multimodal data from OCTA data. The high computational resource requirement of ViT-2SPN proposed by Mohammadreza Saraei et al., and the fact that the self-supervised pre-training phase of ViT-2SPN relies on learning feature representations by generating doubly augmented views, may affect the technique’s robustness in practical applications. Xu YongLi et al. [22] utilized multimodal data from fundus color photographs and OCT to design a method to predict the depth of the residual regression neural network for retinal nerve fiber layer thickness. Despite significant progress in multimodal learning for ophthalmology, there are still some challenges, such as inter-modal misalignment and the handling of noisy data, among other factors [23]. Although multimodal fusion technology significantly improves the diagnostic performance of ophthalmic diseases through cross-modal complementarity, existing methods still have problems regarding the collaborative modeling of heterogeneous data; for example, there are spatial-semantic differences between the two-dimensional wide-area structure of fundus color photographs and the three-dimensional hierarchical features of OCT, and it is difficult to achieve precise alignment of the intermodal information structure by traditional feature splicing or average pooling. In order to break through this limitation, the present study innovatively integrates the attention mechanism into the multimodal feature extraction branch. The attention mechanism is a computerized mechanism that mimics human visual attention, which gives different weights to different parts of the input information, allowing the model to process information more efficiently [24]. In computer vision tasks, by introducing the attention mechanism [25], deep learning models are able to focus better on important regions in the image. Haoran Wang et al. [26] proposed the use of spatial attention, which is widely employed. Ozan Oktay et al. [27] proposed employing attention gates, which can help the model to focus on the information regarding channels in the visual task. Jie Hu et al. [28] proposed using SENet, which enables the network to adaptively focus on the relevant features and suppress non-relevant features, thereby improving the performance of the model. In order to be able to acquire effective feature information, both in the channel and spatially, Sanghyun Woo et al. [29] proposed a CBAM (convolutional block attention module) by integrating channel attention and spatial attention. However, although CBAM enhances feature representation through channel and spatial attention mechanisms, it still fails to adequately capture contextual information from the understanding of complex scenes such as medical images. Transformer’s self-attention mechanism [30] has made a major splash in recent years, and some variants have been derived as well [31,32]. However, compared to convolutional networks, Transformer often needs to be based on a large amount of training data to achieve good results [33,34]. In addition, in some medical fields such as ophthalmology, the available datasets tend to be small in size, and Transformer may not be able to fully utilize it for the best performance. In multimodal learning, the attention mechanism enables deep learning models to extract and fuse feature information from different modalities more efficiently [35]; therefore, for the diagnostic classification of small datasets with category imbalance, a suitable combination of the attention mechanism and a deep learning network is needed to deal with this problem. A convolutional neural network is a more suitable object than Transformer, but, with the continuous development of convolutional neural networks, the depth of the network can cause degradation, making model training difficult. Therefore, Kaiming He et al. [36] proposed a deep residual network ResNet, which solves the problem of network degradation by adding constant mappings between the layers of the convolutional network and constant mappings of the inputs with their nonlinear transformation superposition, solving the network degradation problem. Subsequent ResNet variants have been proposed, such as ResNeXt [37] and Res2Net [38], all of which partially improved the original network. X Liu et al. [39] proposed a ResNet-based deep learning model for detecting diabetic retinopathy from fundus images. Puchaicela-Lozano et al. [40] proposed a structure combining R-CNN and ResNet-50 for image segmentation, in order to detect glaucoma from fundus images. Although ResNet is more suitable than Transformer for certain small-sample medical data processing techniques [41], it may be difficult to accurately localize the important regions in the image with complex ophthalmic images, and it is not possible to fully extract and utilize the detailed features found in the image. Therefore, integrating suitable attention mechanisms into ResNet to strengthen spatial and channel features and enhance the model’s ability to extract global information from ophthalmic images is necessary for effective ophthalmic diagnosis classification models.

## 3. Materials and Methods

In this section, the MOD-Net processing flow is first summarized, and then the design details of specific modules are given.

### 3.1. Overall Architecture

As shown in Figure 2a, the model receives two types of modalities, which are 2D images (fundus color photographs with OCTA 2D en-face retinal pictures) and 3D volume data (OCT and OCTA), which are sliced into 2D slices before being fed into the multi-channel attention-based feature extraction branching network before feature extraction. In the initial stage, Stage 0, the image first undergoes a depth-separable convolution, which significantly reduces the computational effort by decomposing the standard convolution operation, highlighting the important features while suppressing the redundant noise information, and then it undergoes a maximum pooling layer before finally being fed into the residual block.

As shown in Figure 2b, Stages 1–4 consist of multiple residual blocks, based on the frequency-domain transformed attention. The number of residual blocks adopts a 3-4-6-3 hierarchical design, which can balance feature expression capability and computational efficiency and will adapt to the characteristic needs of OCT/OCTA analysis. Multi-scale modeling from the local details of retinal layer boundaries to global semantics is achieved by incrementing the number of residual blocks step by step, and the extraction of medium-complexity features is strengthened in the middle stage (4 blocks in Stage 2), while high-dimensional features are compressed in the end stage (3 blocks in Stage 4) to meet the requirements of DFF feature fusion. The residual structure can be expressed as follows:(1)$$y_{1}=h\left(x_{l}\right)+F\left(x_{l}{,\;W}_{l}\right)$$(2)$$x_{l+1}=f\left(y_{1}\right)$$

$x_{l+1}$ and $x_{l}$ denote the *l* + 1 and *l* inputs of the residual unit, respectively, $F\left(x_{l},\;W_{l}\right)$ is the residual mapping to be learned, W is the learning parameter, $f\left(y_{1}\right)$ is the ReLU function, and $h\left(x_{l}\right)$ denotes the constant mapping. The recursive operation is continued from Equations (1) and (2) to obtain the feature representation of the residual unit l as:(3)$$x_{L}{=x}_{l}+\overset{L-1}{\underset{i=l}{\sum }}F\left(x_{i}{,\;W}_{i}\right)$$(4)$$\frac{\partial loss}{\partial x_{l}}=\frac{\partial loss}{\partial x_{L}}\cdot \frac{\partial x_{L}}{\partial x_{l}}=\frac{\partial loss}{\partial x_{L}}\cdot \left(1+\frac{\partial loss}{\partial x_{L}}\overset{L-1}{\underset{i=l}{\sum }}F\left(x_{i}{,\;W}_{i}\right)\right)$$

$\frac{\partial loss}{\partial x_{L}}$ denotes the gradient of the loss function, where the gradient of the L layer can be passed to those layers shallower than it, while $\frac{\partial loss}{\partial x_{L}}\overset{L-1}{\underset{i=l}{\sum }}F\left(x_{i},\;W_{i}\right)$ denotes the residual gradient and is 1. The problem of information loss during network learning is avoided by propagating the gradient through the layers with weights. These residual blocks help to further extract and optimize the 2D and 3D features using spatial attention, channel attention and frequency-domain transform convolutional attention in the residual blocks for additional feature information extraction and optimization, efficiently utilizing the data, which improves the performance of classification diagnosis in small-sample data scenarios.

After feature extraction, the feature map is fed into the feature fusion module DFF, which consists of the fusion attention gate, spatial attention, and channel attention. The fusion attention gate is realized by the multi-scale attention mechanism, which integrates the features extracted from 2D and 3D branch paths. For integration, after feature fusion, channel attention is used to further refine the feature representation, which can improve the discriminative ability of the network by emphasizing the important feature channels and suppressing the unimportant ones, while spatial attention is used to enhance the model’s perception of the important spatial locations in the image, which is conducive to the network’s focusing on the key regions in the image.

The traditional cross-entropy loss function performs poorly under the small-sample multi-classification task [42], so the current method adopts Facol Loss [43] as the loss function, which alleviates the sample imbalance problem by dynamically increasing the weights of difficult-to-categorize samples and decreasing the weights of easy-to-categorize samples so that the model focuses on the training of difficult-to-train samples. As shown in Equations (5) and (6), *i* denotes the *i*-th sample and γ denotes the parameter controlling the weight of the category.(5)$$L=-\overset{N}{\underset{i=1}{\sum }}{\left({1-p}_{i}^{t}\right)}^{\gamma }\log\left(p_{i}^{t}\right)$$(6)$$p_{i}^{t}=\{\begin{array}{c} p_{i}{,\;y}_{i}=1\\ {1-p}_{i},\;otherwise \end{array}$$

### 3.2. FTCA Module

Ocular images naturally contain both 2D and 3D modal information; 2D images (fundus photographs) provide planar structural and textural features, while 3D somatic data (OCT/OCTA) reflect the hierarchical structure of ocular tissues. They both contain rich textural and structural information, e.g., fundus color photographs capture the morphological features of the optic disc, macula, and vascular system, while the B-scan sequence of OCT constitutes the axial resolution of the retinal structure. Therefore, the extraction and enhancement of these features are crucial for subsequent classification. However, existing methods of attentional mechanisms usually focus only on the channel domain or the spatial domain, thereby lacking an effective attentional mechanism dedicated to grayscale images. During the study, it was found that frequency domain information can effectively capture the global structure and local details of the image, so this study proposes a comprehensive and efficient attention mechanism that takes into account the channel, spatial, and frequency domains. The channel attention establishes the mapping relationship between color-photographed blood vessel morphology and OCT blood flow signals. The spatial attention adapts the axial hierarchical structure of the OCT and the OCTA, and the frequency-domain attention transforms the feature mapping to the spectral domain for global manipulation, realizing a full range of ocular image sensory fields.

Fast Fourier transform (FFT) [44], which plays an important role in frequency domain analysis and filtering, can be used to perform the frequency domain filtering of images and can convert images from the spatial domain to the frequency domain so that the high-frequency and low-frequency information is separated in order to resolve and recognize different structural features in the image. Inspired by the work of Lu Chi et al. [45], this study proposes a channel-space hybrid-attention FTCA based on frequency domain transformation, as shown in Figure 2b, which can focus on the feature information in the channel, spatial, and frequency domains simultaneously to optimize the computational process and better understand the image content when processing grayscale images, which helps MOD-Net to capture richer contextual information. This multiplies the output features of the channel attention module and spatial attention module and also the fast Fourier transform element by element, to obtain the final attention-enhanced features, which are used as inputs to the subsequent network layers to suppress noise and irrelevant information while retaining key information.

Specifically, the frequency domain attention consists of two interconnected paths, one in the spatial and the other in the frequency domain, each of which captures feature information from the other and complements each other. This information exchange is performed internally, first using an FFT to the frequency domain, then via a convolution operation in the frequency domain to capture the global features, and finally by using an inverse transform back to the spatial domain. The internal update process is shown in Equations (7) and (8):(7)$$Y^{l}{=Y}^{l\to l\;}{+Y}^{g\to l}{=f}_{l}\left(X^{l}\right){+f}_{g\to l}\left(X^{g}\right)$$(8)$$Y^{g}{=Y}^{g\to g}{+Y}^{l\to g}{=f}_{g}\left(X^{g}\right){+f}_{l\to g}\left(X^{l}\right)$$

In channel attention, adaptive filtering is performed on input features in the channel dimension with constant channel dimension and compressed spatial dimension. The feature map, with an input size of C × H × W, is passed through a parallel MaxPool layer and AvgPool layer and is then passed through an MLP layer and output. These two outputs are then added element by element to form a channel-weighting module, and then a Sigmoid activation function is passed to obtain the output of the channel attention. This output is then multiplied by the original feature map and changed back to the size of C × H × W. The process is shown in Equations (9) and (10), where σ denotes the Sigmoid activation function and $F^{\prime }$ denotes the processed features.(9)$${Attention}_{c}\left(F\right)=\sigma \left(MLP\left(AvgPool\left(F\right)\right)+MLP\left(MaxPool\left(F\right)\right)\right)=\sigma \left(W_{1}\left(W_{0}\left(F_{avg}^{c}\right)\right){+W}_{1}\left(W_{0}\left(F_{max}^{c}\right)\right)\right)$$(10)$$F^{\prime }=Attention_{c}(F)\times F$$

In spatial attention, the input features in the spatial dimension are adaptively filtered with a constant spatial dimension and compressed channel dimension. The output of the channel attention is passed through the maximum pooling layer and the average pooling layer to obtain two 1 × H × W feature maps; the two feature maps are then spliced by the concat operation and the final feature maps are obtained after Sigmoid. Finally, the output is multiplied by the original maps to change it back to the C × H × W size. The same process is shown in Equations (11) and (12).(11)$${Attention}_{s}(F^{\prime })=\sigma (f([AvgPool(F^{\prime });MaxPool(F^{\prime })]))=\sigma (f([{F^{\prime }}_{avg}^{s};{F^{\prime }}_{max}^{s}]))$$
(12)$$F^{\prime \prime }=Attention(F^{\prime })\times F^{\prime }$$

### 3.3. DFF Module

There are some problems with traditional feature fusion methods such as the concat operation method. As the dimensionality of the features increases, the complexity of the model increases, which may lead to overfitting, especially if the amount of training data is insufficient. Also, these features may contain redundant information; the concat operation will combine these redundant pieces of information to amplify the noise, making the correlation between the features insufficient, meaning that the features learned by MOD-Net are not guaranteed to be optimal. Moreover, features at different scales contain different levels of information; low-level features are rich in details and edge information, while high-level features contain more contextual information. Establishing how to efficiently utilize the features of the different modalities learned by the multi-modal network is a key issue.

Therefore, in this study, a DFF module consisting of a fusion attention gate, channel attention, and spatial attention is designed, for which the FAG structure is shown in Figure 2c. Firstly, a multi-scale attention mechanism based on grouped convolution is utilized to process the color photo and the OCT, respectively, and the feature maps from different scales are processed through the attention mechanism to enhance the important features, suppressing the irrelevant features to retain the modality-specific features. Afterward, feature-weighted fusion is performed in order to enhance the feature information from different levels. This process is shown in Equation (13):(13)$$FAG\left(image,\;volume\right)=Sigmoid\left({Conv}_{1\times 1}\left(ReLU\left(W_{i}{+W}_{v}\right)\right)\right)$$

This feature-weighted fusion approach enhances the model’s discriminative ability for different regions in the image and avoids the problem of performance degradation due to the data dependency of traditional methods when dealing with high-resolution images, especially in complex medical ophthalmology images, which is crucial for improving the model’s classification accuracy. Then, channel attention (learning cross-modal channel dependencies through MLP with shared weights) and spatial attention (using cross-entropy attention to enhance the response of two-modal co-occurring lesion regions) are sequentially used in tandem to emphasize the relevant features in two dimensions while integrating the information from different scales to enhance the performance of the model. This combination of methods can better capture complex feature relationships while suppressing irrelevant regions, thereby enhancing overall feature representation. Through the above design, the DFF module can effectively utilize the information from both low-level features and high-level features to achieve efficient classification.

## 4. Experiment

The hardware platform for this experiment is the NVIDIA GeForce RTX 4060Ti 16G at 2535 MHz, the deep learning platform used is Pytorch 2.1.0, and the programming language is Python 3.10. The experimental results on two publicly available multimodal datasets, as presented in this section, demonstrate that the proposed method is able to efficiently optimize the feature extraction and has superior diagnostic classification performance.

### 4.1. Datasets

In this study, two multimodal ocular datasets were used to evaluate the performance of the model, namely, the OCTA-500 dataset [46] and the GAMMA dataset [47], each of which focuses on different ocular diseases. These two datasets contain two imaging modalities, respectively, as shown in Figure 3a. The first group uses the OCTA-500 dataset; the first row shows the OCTA 2D en-face retinal images, which clearly show the capillaries and tiny lesions in all layers of the retina, with a higher density of normal vascular capillaries and a smaller range of macular center pits. The second row shows the OCTA B-scan image and the (b) group is of the GAMMA dataset. The first row shows the color photograph of the fundus, which demonstrates the contrast between the blood vessels and the background, and the second row shows the OCT B-scan image, which reflects the structure of each layer of the eye; the thickness of each layer of the normal eye is uniform, and the interlayer structure is intact. The B-scan image is shown with the 3D volume sliced into 2D slices.

The distribution of the respective disease categories of the two datasets is shown in Figure 4a. The OCTA-500 group is divided into three categories, which show normal eyes, age-related macular degeneration, and diabetic retinopathy, totaling 169 groups with 67,769 images. Figure 4b The GAMMA group is divided into three categories, which show normal eyes, early glaucoma, and intermediate and progressive glaucoma, totaling 100 groups with 25,700 images. It can be seen that the ratio of normal cases compared to disease-related cases is about 2:1.

### 4.2. Pre-Processing

Before training, the 2D images and 3D volume data (read in grayscale format) were read using the OpenCV and PIL libraries, respectively. The 2D images, such as the fundus photographs, have more planar information, which provides a visualization of the overall structure and helps to locate the lesion area more quickly. The 3D volume data, such as the OCT images, provide volumetric and depth information, which can demonstrate the layered structure and thickness variations in the ocular tissues. The OCT B-scan and OCTA B-scan data were stored in a 3D NumPy array so that the dimensionality of the array could be adjusted to match the input requirements of MOD-Net. After reading the images, predefined random cropping, flipping, and rotating operations were applied to enhance the diversity of the training data, which ensured the effective loading and enhancement of the data. During the training process, Gaussian noise enhancement was also used to decide whether to add Gaussian noise to the current image or not, with a probability of 60%, and the variance of the noise was randomly selected to be between 0.01 and 0.05.

Convolutional neural networks (CNNs) play an important role in image classification and recognition; however, training CNNs from scratch usually requires a large amount of data and computational resources. In addition, training CNNs using small-sample datasets tends to lead to overfitting, so this method uses transfer learning techniques. The learning effect of the target task is solved using the data features of the source domain $D_{s}=\left\{X_{s},\;P_{s}\left(x\right)\right\}$ and knowledge of the source task $T_{s}=\left\{Y_{s},\;f_{s}\left(\cdot \right)\right\}$ (where X denotes the feature space, P(x) is denoted as the edge distribution, Y denotes the class labeling space, and f(∙) denotes the target prediction function).

### 4.3. Performance Metrics and Experimental Settings

In order to better evaluate the performance of MOD-Net, Accuracy, Precision, F1-Score, Recall, and the Confusion Matrix are used as evaluation metrics, and the larger values of these metrics represent better performance. Among them, Precision, F1-Score, and Recall are weighted types, so that the performance of the multiclassification model in the face of category imbalance can be effectively evaluated. The definitions of these metrics are expressed as:(14)$$Accuracy=\frac{TP+TN}{TP+FP+TN+FN}$$(15)$$Precision=\frac{TP}{TP+FP}$$(16)$$Recall=\frac{TP}{TP+FN}$$(17)$$F1=2\;\times \;\frac{Pre\;\times \;Rec}{Pre+Rec}$$
where TP, FP, TN, and FN are the numbers of true positives, false positives, true negatives, and false negatives, respectively. This method divides the training and test sets according to a ratio of 8:2, and in the training set, it continues to divide the training and validation sets according to the ratio of 8:2. For training, an Adam optimizer with a learning rate of lr = 1 × 10^4^, a batch size of 4 (Batchsize = 4), a maximum number of iterations of 100 (Epoch = 100), and the γ value in Focal loss set to 3. These hyperparameters were chosen during repeated iterations of trial and error to determine the best performance that could be obtained on the validation set, in order to evaluate the test data using these configurations.

### 4.4. Results and Visualization

Based on the abovementioned environment and configuration under which MOD-Net was trained, Figure 5 shows the training loss of the model on two datasets, respectively, where Figure 5a is the result with OCTA-500 and Figure 5b is the result with GAMMA.

The loss curve in (a) shows the trend of the loss value of the model in the training process, as can be seen from the visualization; with an increase in the number of iterations, the fluctuation of the loss value is gradually reduced until it stabilizes, and the loss value shows a decreasing trend as a whole, which indicates that the model is gradually optimized for convergence in the learning process, and its performance has been improved. In the late stage of the training period, the loss value is close to zero, which indicates that the model has achieved a better fitting effect. In order to verify the generalization of MOD-Net, at the same time, the GAMMA dataset is also trained to verify this, as can be seen from (b), although there are obvious fluctuations in the curve. However, in the middle and late stages, the loss value tends to stabilize and remain in the lower range, and the fluctuations occur with a significantly lower frequency, which reflects the model’s ability to generalize.

The training is completed and followed by testing. In order to validate the performance of MOD-Net, 11 representative classification models with excellent performance and the latest SOTA method in the field, GeCoM-Net, and ViT-2SPN were selected for comparison testing in this paper. These are: VGG [48], ResNet [31], EfficientNet [49], Swin Transformer [50], DaViT [51], RepLKNet [52], VAN [53], ConvNeXt V2 [54], HiViT [55], GeCoM-Net [21], and ViT-2SPN [15]. In the field of deep learning, VGG employs consecutive convolutional layers and small convolutional kernels to enhance the feature extraction capability and improve the performance of the model. ResNet introduces the concept of residual learning, which significantly improves the depth and performance of the network by jumping connections. EfficientNet optimizes the efficiency and accuracy of the model with a composite scaling strategy and achieves high performance while maintaining a small scale. Swin Transformer applies the advantages of Transformer to visual tasks by introducing a hierarchical window attention mechanism. DaViT improves feature representation in visual tasks by combining convolution and self-attention mechanisms. RepLKNet uses a large convolutional kernel and reparameterization to improve the network’s ability to learn sensory field and shape information while maintaining computational efficiency. VAN focuses on enhancing the selectivity of visual features through attentional mechanisms and hierarchical feature extraction. ConvNeXt V2 efficiently combines the strengths of the convolutional network and Transformer architectures. HiViT simplifies the hierarchical design of Swin Transformer’s hierarchical design and improves processing efficiency through hierarchical patch embedding, which is especially good when processing high-resolution images. GeCoM-Net is a multimodal fusion network for the efficient diagnosis of ophthalmic diseases. ViT-2SPN achieves retinal OCT classification using dual-stream self-supervised pre-training with the Vision Transformer framework. Table 1 shows the results of the comparison test.

From the table, it can be seen that MOD-Net outperforms the other methods using the OCTA-500 dataset, and also outperforms the SOTA method GeCoM-Net, with an accuracy of 0.7407, a precision of 0.8049, a weighted F1 score of 0.7445, and a recall of 0.7407, which is significantly better than the other compared methods. Due to the integration of frequency domain attention, channel attention, and spatial attention in FTCA, which significantly improves the detection rate of tiny lesions, MOD-Net shows superior performance when processing the OCTA-500 task. Other methods, such as DaViT, which is a network with Transformer architecture, perform poorly. ResNet and EfficientNet, although having an accuracy of 0.7037 and showing strong performance, still show some gaps compared to MOD-Net, and traditional convolutional networks (ResNet/EfficientNet) are not as efficient as the conventional networks, due to the fixed scaled receptive fields exhibiting a limited performance in such tasks. The performance of RepLKNet is within the range between traditional convolutional networks and Transformer-like networks, while GeCoM-Net still shows a gap.

The lightweight design of MOD-Net avoids the overfitting problem of Transformer-like models (ViT-2SPN/Swin) with limited medical data, which makes MOD-Net more relevant to ophthalmology. Since the FTCA designed for OCT/OCTA medical grayscale images focuses on feature information in the channel, spatial, and frequency domains at the same time, it optimizes the computational process to better understand the image content and helps MOD-Net to capture richer contextual information. Meanwhile, FAG utilizes the grouped convolution-based multi-scale attention mechanism to process the feature maps at different scales, in order to enhance important features and inhibit the irrelevant features, and then performs feature-weighted fusion to enhance the feature information at different levels; finally, it enhances the model’s ability to discriminate between different regions in the image. These advantages make MOD-Net more suitable for ophthalmology diagnosis, and it achieves better performance results. In order to further visualize the performance, a confusion matrix is drawn up in this paper for visualization and comparison, as shown in Figure 6.

The confusion matrix shows the relationship between model predictions and actual labels, with the numbers on the diagonal indicating the number of correctly categorized samples and the numbers off the diagonal indicating the number of samples that were misclassified. From the confusion matrix in Figure 6, it can be visualized that MOD-Net has a lower number of misclassifications for the AMD and DR categories, showing a better feature recognition ability on the case category. Although this is not the highest number of correct classifications for the normal category, its overall performance is more stable in dealing with the AMD and DR categories, and, thus, is important for use in early clinical diagnosis. In addition, we tested the model on the GAMMA dataset.

According to the results in Table 2, it can be seen that MOD-Net also shows the best performance on the GAMMA dataset, which is not only reflected in the model’s accuracy but also in the lower number of misclassifications for the categories of early and middle–late stages, which suggests that MOD-Net is able to accurately capture the intrinsic features of the data and make accurate predictions when facing brand-new category-imbalanced small-sample data. The model complexity and generalization ability are balanced by a well-designed network structure and pre-processing to avoid overfitting. In the clinical environment, deep learning models need to deal with a variety of different cases, and MOD-Net’s generalization ability ensures that it can provide more accurate diagnostic assistance in complex situations, proving that MOD-Net has great potential for practical applications in ophthalmic medical image processing.

Finally, the heat maps generated by MOD-Net and other deep learning models on the two datasets (observing on which region the network focuses more) for interpreting the decision-making process of the models are shown in Figure 7, with group (a) showing the heat maps of some models on the OCTA 2D en-face images and group (b) showing the heat maps of some models on the retinal fundus color photographs. The OCTA 2D en-face group provides high-resolution images of retinal vascular tissue [56], so it is important to view and interpret the en-face images [57]. As can be seen in group (a), the present method overlays the extracted features from the inside to the outside in a regular circle, which takes into account the foveal avascular zone (FAZ) and the peripheral vasculature. Thus, the OCTA-500 dataset achieved good performance, while EfficientNet-B3 mainly focused on the left side of the FAZ, RepLKNet only focused on a small area above the FAZ, and HiViT had a wide range but a very small feature extraction area. In short, these models were distracted from identifying key features. The initial clinical diagnosis of glaucoma mainly lies in the optic cup and optic disc and their ratios [58], as can be seen in group (b). The present method achieved good performance in the GAMMA dataset by focusing mainly on the optic cup and optic disc regions for coverage of the extracted features, while ResNet50 focused on the lower leopard-shaped fundus region, which can only support a diagnosis of myopia and does not offer full support of the diagnosis of glaucoma [59], while Swin Transformer and DaViT analyses are too wide to fully focus on the target region, so other features may interfere with the diagnostic classification results.

In conclusion, the heat map shows that MOD-Net has high accuracy in identifying key features, with better generalization ability and recognition accuracy in processing the dataset, which is especially important for ophthalmic medical image analysis, making it more effective in clinics when playing the role of disease diagnosis assistance.

### 4.5. Ablation Experiments

In this paper, an ablation study was carried out, which was consistent except for the model structure, to validate the effectiveness of the FTCA and DFF modules in multimodal learning networks.

As shown in Table 3, MOD-Net 0, based on FTCA, only achieved an accuracy of 0.7037 with the OCTA-500 dataset, reflecting the effectiveness of the hybrid attention model based on frequency domain transformation. MOD-Net 1, based on DFF, only achieved an accuracy of 0.6667 with the OCTA-500 dataset, reflecting the effectiveness of the multi-scale attention fusion module. They both exceeded the performance of most classification models. MOD-Net, based on both FTCA and DFF, achieved the best performance. In conclusion, the results of the ablation experiments intuitively show that the method proposed in this paper can effectively improve the performance of symmetry-aware dual-stream networks.

## 5. Conclusions

In this study, we propose a dual-branch network with hybrid attention, MOD-Net, to perform the diagnostic classification of common eye diseases. MOD-Net adopts a residual block of hybrid attention based on frequency domain transform for multimodal feature extraction, while FTCA performs the attention computation on the channel, spatial, and frequency domains, respectively. This enhances the most important spatial and channel features, transforms the traditional spatial domain into the frequency domain, and enhances the information in the frequency domain using a fast Fourier transform to increase the efficiency of the feature computation, thereby solving the problem of the inefficiency of feature extraction shown by the original network on the 2D modal and 3D modal data, with low efficiency of feature extraction. In the feature fusion stage, the DFF module, which is based on a multi-scale attention mechanism, is used to effectively fuse features from different levels and emphasize the task-related channel and spatial information in the feature map, which improves the model’s diagnostic classification accuracy by integrating information from different scales. Finally, experiments on the publicly available OCTA-500 and GAMMA datasets demonstrate that MOD-Net is suitable for multi-category imbalanced small-sample ophthalmology datasets, and the results outperformed the existing methods.

## Figures and Tables

**Figure 1 bioengineering-12-00565-f001:**
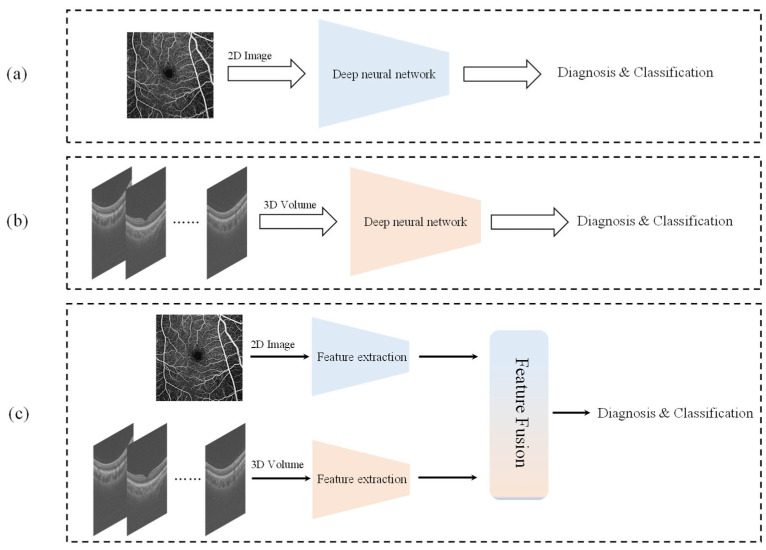
A brief comparison between traditional unimodal networks and the multimodal networks used in this study: (**a**) a deep learning network for 2D data, (**b**) a deep learning network for 3D data, and (**c**) a deep learning network that simultaneously extracts features from 2D and 3D modal data and then performs feature fusion.

**Figure 2 bioengineering-12-00565-f002:**
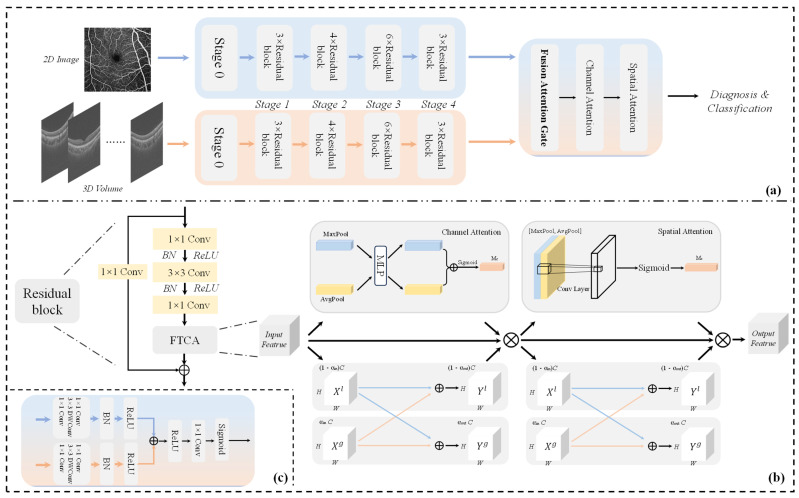
Mod-NET processing flow: (**a**) shows the overall MOD-Net architecture, while (**b**) shows the residual block structure. The dual-feature fusion (DFF) module consists of a fusion attention gate with channel attention and spatial attention. Insert (**c**) shows the fusion attention gate module.

**Figure 3 bioengineering-12-00565-f003:**
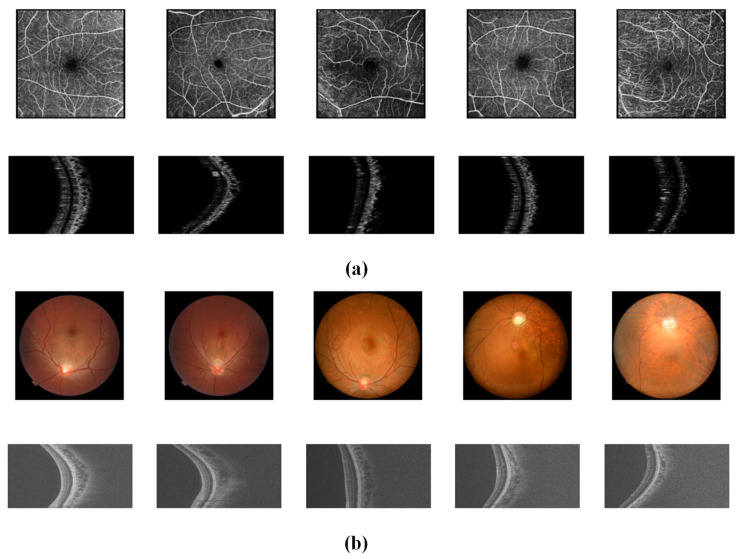
Visualization of the two datasets: (**a**) visualization for the OCTA-500 dataset, (**b**) visualization for the GAMMA dataset.

**Figure 4 bioengineering-12-00565-f004:**
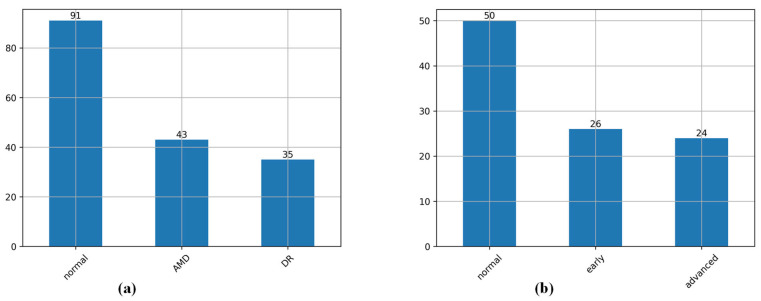
Histogram of category distribution: (**a**) the distribution of eye disease categories for the OCTA-500 dataset, and (**b**) the distribution of eye disease categories for the GAMMA dataset.

**Figure 5 bioengineering-12-00565-f005:**
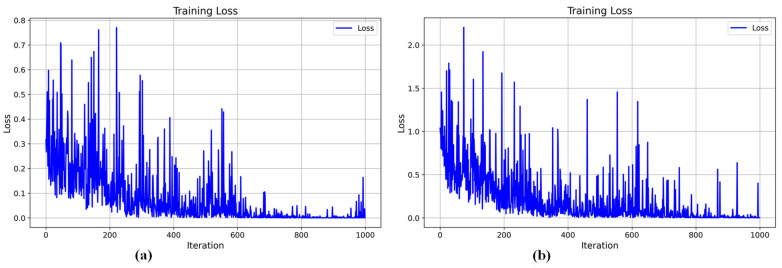
Real-time visualization of the loss function trend during training: (**a**) is the loss curve of OCTA-500, and (**b**) is the loss curve of GAMMA.

**Figure 6 bioengineering-12-00565-f006:**
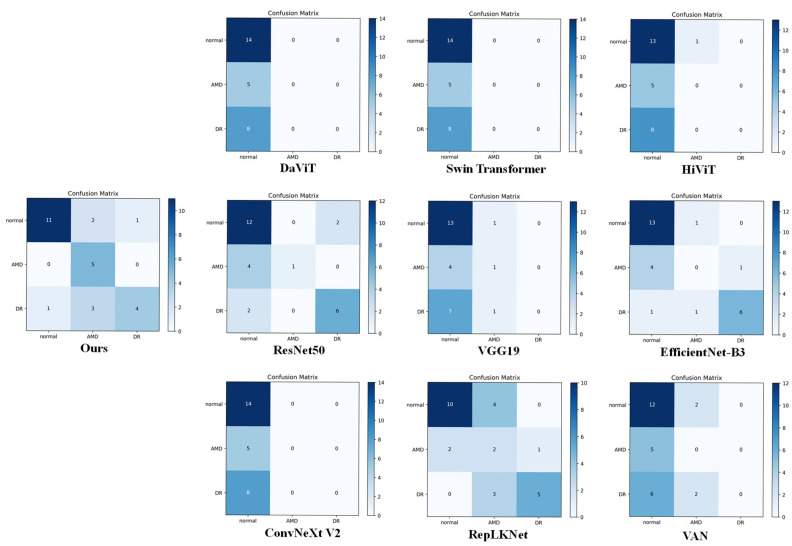
Confusion matrix comparison using the OCTA-500 dataset.

**Figure 7 bioengineering-12-00565-f007:**
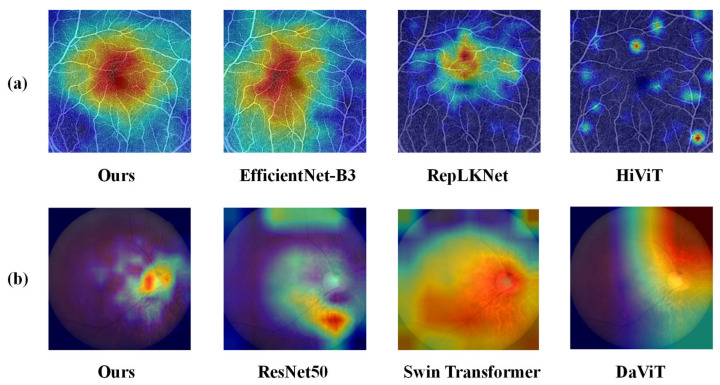
Comparison of heat maps on the two datasets: (**a**) is a heatmap on the OCTA-500 dataset, while (**b**) is a heatmap on the GAMMA dataset.

**Table 1 bioengineering-12-00565-t001:** Comparison with SOTA methods using the OCTA-500 dataset. **↑** means the higher the indicator, the better.

Method	Author	Accuracy (↑)	Precision (↑)	F1_weighted_ (↑)	Recall (↑)
VGG	Simonyan, K et al.	0.5185	0.3426	0.4011	0.5185
ResNet	He, K et al.	0.7037	0.7530	0.6728	0.7037
EfficientNet	Tan, Mingxing et al.	0.7037	0.6285	0.6583	0.7037
Swin Transformer	Z. Liu et al.	0.5185	0.2689	0.3541	0.5185
DaViT	Ding, Mingyu et al.	0.5185	0.2689	0.3541	0.5185
RepLKNet	Ding, Xiaohan et al.	0.6296	0.7202	0.6634	0.6296
VAN	Guo, Meng-Hao et al.	0.4444	0.2705	0.3363	0.4444
ConvNeXt V2	Woo, S et al.	0.5185	0.2689	0.3541	0.5185
HiViT	Zhang, Xiaosong et al.	0.4815	0.2593	0.3370	0.4815
GeCoM-Net	Wang, Y et al.	0.7241	0.7786	0.6936	0.7241
ViT-2SPN	Saraei, Mohammadreza et al.	0.7293	0.7913	0.7225	0.7293
MOD-Net	Our study	0.7407	0.8049	0.7445	0.7407

**Table 2 bioengineering-12-00565-t002:** Comparison with SOTA methods using the GAMMA dataset. **↑** means the higher the indicator, the better.

Method	Author	Accuracy (↑)	Precision (↑)	F1_weighted_ (↑)	Recall (↑)
VGG	Simonyan, K et al.	0.4500	0.3281	0.3403	0.3500
ResNet	He, K et al.	0.4000	0.2800	0.2927	0.4000
EfficientNet	Tan, Mingxing et al.	0.5000	0.6633	0.4302	0.5000
Swin Transformer	Liu, Z et al.	0.3500	0.1225	0.1815	0.3500
DaViT	Ding, Mingyu et al.	0.3500	0.1225	0.1815	0.3500
RepLKNet	Ding, Xiaohan et al.	0.4500	0.3801	0.3750	0.4500
VAN	Guo, Meng-Hao et al.	0.4000	0.4861	0.2835	0.4000
ConvNeXt V2	Woo, S et al.	0.3500	0.1225	0.1815	0.3500
HiViT	Zhang, Xiaosong et al.	0.4500	0.3775	0.3442	0.3775
GeCoM-Net	Wang, Y et al.	0.7035	0.7245	0.6882	0.7035
ViT-2SPN	Saraei, Mohammadreza et al.	0.7200	0.7333	0.7015	0.7200
MOD-Net	Our study	0.7500	0.7500	0.7400	0.7500

**Table 3 bioengineering-12-00565-t003:** Ablation experiment results with the OCTA-500 dataset. **↑** means the higher the indicator, the better.

Method	FTCA	DFF	Accuracy (↑)	Precision (↑)	F1_weighted_ (↑)	Recall (↑)
MOD-Net 0	✓		0.7037	0.7615	0.7076	0.7037
MOD-Net 1		✓	0.6667	0.7333	0.6673	0.6667
MOD-Net	✓	✓	0.7407	0.8049	0.7445	0.7407

## Data Availability

Data will be made available on request.

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
