# Peer review of "Dual-Branch Network with Hybrid Attention for Multimodal Ophthalmic Diagnosis"

_bioengineering, 2025, doi:10.3390/bioengineering12060565_

Round 1

Reviewer 1 Report

Comments and Suggestions for Authors

This paper presents a dual-branch network with hybrid attention for multimodal ophthalmic diagnosis. The structure of the paper is fine. The authors presented all the sections carefully and in a good way. When the introduction section is checked, the authors used 13 references but reference number 7 is missing and the information about these references are not given in details.  Another example is that "Studies have shown" is given line 79 but there is only 1 reference paper given on this line. Figure 1 is taken from reference number 12 but it is not mentioned in the figure caption. The authors should check the paper carefully.

Reviewer 2 Report

Comments and Suggestions for Authors

The article presents Dual-Branch Network with Hybrid Attention for Multimodal Ophthalmic Diagnosis. The article covers the need and the motivation for the research with related work and the discussion on the proposed methods followed by the results and discussion. Some points need to be addressed.

  • The literature review section needs to be revised. Recent and relevant articles need to be included. The recent articles included are review articles. The authors need to include a comparative analysis of the existing methods. The outcome of the literature review needs to be included at the end of the section.
  • Part (a) in Figure 1 consists of duplicate components. The only difference between the two is the slicing of the 3D into 2D as stated in line 181. The figure can be redrawn by removing the duplicate components. Discussion of the number of residual blocks and the fusion attention gate also need to be discussed.
  • The comparative analysis needs to be revised. The authors have compared the proposed model with generic methods. The methods compared are not even included in the literature review section. The methods are also old. The authors need to include recent specialized methods for comparison. The results reported in original studies have achieved better results on other datasets. How the methods were tuned for the dataset of this study needs to be included. The articles that published the datasets have also reported better results in terms of accuracy.
  • The authors have presented the results in detail however the discussion covers general statements that the proposed model achieved improvements over the compared methods. The discussion on the rationale behind the improvements is missing.

Reviewer 3 Report

Comments and Suggestions for Authors

The authors here propose a dual-branch deep learning network with hybrid attention to 
perform diagnostic classification of common eye diseases. 

Outstanding work, supported by the excellent manuscript. The introduction is very informative, targeting a more technical audience and less clinical. Global and critical description of the related work. The presented method, based on a dual-branch network, as well as the experimental part, is clear and thoroughly described.  The comparisons provided in the first two tables are impressive, supporting the third one.

Finally, the results of the ablation experiments intuitively show that the method proposed in this paper can effectively improve the performance of the symmetry-aware dual-stream network

Author Response

Thank you so much for taking the time to review the paper.

Comments 1: [The introduction is very informative, targeting a more technical audience and less clinical.]

Response 1: [Thank you, this is mainly due to the fact that the research team is mainly technical in origin, and we will pay attention to this in our subsequent work at the intersection of AI and medicine by interacting more with clinical experts and reflecting it in paper.]

Lastly, thank you very much once again for recognizing this article.

Reviewer 4 Report

Comments and Suggestions for Authors

The paper proposes dual dual-branch neural network enhanced by attention modules for medical applications. In general, the idea is interesting but the paper is not ready for publication. My aspects should be improved in the current state of the paper like:
-not sure if the background in first two sections are valid, the paper shows research that should be evaluated in 2025, but that is not what i found in the first two sections
-fig 1 is too general
-section 3 with methodology does not explain the architecture and its justification is missing. I mean, why such a layer is now, and not after some other type etc. The evaluation of processed data in the model is not given.
-The medical application is just given, what about a formal model of such software and architecture? Check solutions like a medical image system with dual-input CNN 
-The explainability of the model is also not shown
-how did you choose the attention module?
-comparison with state of art in tab 1 and 2 should describe mainly the last 2 years, but there is only 1 such reference, so I cannot agree that the results are valid in 2025
-also, the authors send paper to the system in 2025, but the paper explains that the model was valid in 2024, not in 2025 (check tables and references)
-ablation studies are missing

Round 2

Reviewer 1 Report

Comments and Suggestions for Authors

The authors updated the paper accordingly

Reviewer 2 Report

Comments and Suggestions for Authors

Most of the concerns have been addressed. The related work section still lacks the comparative analysis.  

Reviewer 4 Report

Comments and Suggestions for Authors

The paper can be accepted